# TruD technology for the study of epi- and endothelial tubes *in vitro*

**Steen H. Hansen**  *

Department of Pediatrics, Division of Gastroenterology, GI Cell Biology Laboratory, Hepatology and Nutrition, Boston Children's Hospital, Harvard Medical School, Boston, Massachusetts, United States of America

* steen.hansen@childrens.harvard.edu

## Abstract

Beyond the smallest organisms, animals rely on tubes to transport cells, oxygen, nutrients, waste products, and a great variety of secretions. The cardiovascular system, lungs, gastrointestinal and genitourinary tracts, as well as major exocrine glands, are all composed of tubes. Paradoxically, despite their ubiquitous importance, most existing devices designed to study tubes are relatively complex to manufacture and/or utilize. The present work describes a simple method for generating tubes *in vitro* using nothing more than a low-cost 3D printer along with general lab supplies. The technology is termed "TruD", an acronym for <u>tru</u>e <u>d</u>imensional. Using this technology, it is readily feasible to cast tubes embedded in ECM with easy access to the lumen. The design is modular to permit more complex tube arrangements and to sustain flow. Importantly, by virtue of its simplicity, TruD technology enables typical molecular cell biology experiments where multiple conditions are assayed in replicate.

**Data Availability Statement:** All relevant data are within the manuscript and its Supporting information files.

**Funding:** The author(s) received no specific funding for this work.

## Introduction

The majority of our internal organs are comprised of interconnecting tubes arranged in parallel and/or serial circuits of varying complexity. As such, tubes are essential for all forms of life from worms to humans and widely implicated in disease. Tubes are integral to the circulation and lungs and exert numerous functions in distributing cells, oxygen, metabolites, immune and stem cells. The GI tract is the longest tube in the human body and serves dual functions in combining nutrient intake and excretion of waste. Furthermore, like other mucosal surfaces, the GI tract performs a pivotal role in innate and adaptive immune sensing of the environment. Equally essential functions are mediated by the female and male genitourinary tracts. Moreover, all exocrine glands are completely reliant on tubes of varying length to traffic secretions to the inner and outer surfaces of the body where they are needed for homeostasis of the individual in which they reside, or in the case of the mammary gland, for nutrition of offspring [1–4]. Thus, there is no complex life without a plethora of tubes and yet, in a time of great technological sophistication, there is limited access for the majority of life scientists to study epi- and endothelial tubes *in vitro*.

In recent years, reports have emerged from bioengineering labs describing ingenuous and sophisticated devices for generating epi- and/or endothelial tubes [5–16]. These devices are groundbreaking in terms of permitting studies of tubes *in vitro* that previously could not be

**Competing interests:** The author has declared that no competing interests exists.

undertaken. As such, they are highly useful for the tasks for which they were engineered. However, most of these devices require special tooling to manufacture and are not readily modifiable to be used in other contexts. The vast majority of studies of epi- and endothelial cells are conducted by scientists without bioengineering expertise for whom an unmet need exists for incorporating tubes in their work. The present study seeks to fill this void. Accordingly, a simple system is described herein; one that allows literally anyone to cast and study epi- and endothelial tubes *in vitro*. This system is coined "TruD", an acronym for <u>tru</u>e <u>d</u>imensional, as both Transwells and cysts/organoids cultured in ECM are three dimensional but not "true" to the shape of the tubular structures they often are supposed to mimic.

## Materials and methods

### Chemicals and reagents

FCS (R&D Systems), VascuLife® VEGF-Mv Endothelial Complete Kit (Lifeline Cell Technology cat no. SKU-LL0005), DMEM (Gibco, cat. no. 11965092), Penicillin-Streptomycin (5,000U/ml; Gibco, cat. no.15070063), Fugene 6 (Promega cat. no. 2691), Polybrene (Santa Cruz Biotechnology, cat. no. sc-134220), 100 mm dishes (GenClone, cat. no. 25–202), 6-well dishes (Cellstar, cat.no. 657 160), 1 ml syringes (BD, cat. no. 309659), 20G x $1^1/_2$ needles (BD, cat. no. 305176), 22G x $1^1/_2$ needles (BD, cat. no. 305156), 27G x $1^1/_2$ needles (BD, cat. no. 301629), 0.45 µm filters (Corning, cat. no. 431220); 12 mm coverslips (Electron Microscopy Sciences, cat. no. 7229002); Blue Fluorescent Microspheres 1.3g/cc 1-5um (Cospheric, cat. no. FMB-1.3 1-5um); Orange-Yellow Fluorescent Microspheres 1.3g/cc 1-5um (cat. no. FMOY-1.3 1-5um); Alexa Fluor™ 594 Phalloidin (Molecular Probes, cat. no. A12381); DAPI solution (1 mg/ml; Thermo Scientific, cat. no. 62248); PLA Plus 1.75 mm 3D printing filament (Duramic).

### Cell culture and lentiviral transduction

Madin-Darby Canine Kidney strain II (MDCK) cells, TR-T10 MDCK cells expressing TetR (TR-T10) [17], MDA-MB-231 and 293T cells, all lab stock, were cultured in DMEM with 10% FCS. Telomerase-immortalized human aortic endothelial cells (Telo-HAECs) were propagated in complete Vasculife Endothelial Cell Medium. 293T cells were split 1:10 two times per week, while MDCK, TR-T10, Telo-HAEC and MDA-MB-231 cell lines were split 1:10 once a week. For experimentation, culture medium was supplemented with pen-strep.

To derive MDCK and Telo-HAEC cells expressing mEGFP or TR-T10 cells and MDA-MB-231 expressing mCherry, 60–70% confluent 293T cells, cultured in a 100 mm dish with 5-ml medium, were transfected with 2 µg each of transfer vector, psPAX2 (Addgene, Plasmid #12260), and pCMV-VSV-G (Addgene, Plasmid #8454) and 18 µl of Fugene 6 mixed in 1 ml of Optimem for 30 minutes prior to addition to 293T cells. Transfer vectors were the following: empty FgH1tUTG encoding EGFP (Addgene, Plasmid #70183) and empty pUltra-Hot encoding mCherry (Addgene, Plasmid #24130). Twenty-four hours later, the medium was replenished and forty-eight hours later, medium supplemented with 5 µg/ml polybrene was harvested with a syringe, passed through a 0.45 µm syringe filter, and added to the target cells. This process was repeated twice after which virtually all cells expressed EGFP or mCherry.

### 3D printing of TruD chips and support materials

TruD chips and support materials were printed on a Prusa MK3S printer with a 0.4 mm nozzle using PLA+ filament and standard settings: 220/60˚C nozzle/bed temperature; 100% infill; 0.15 mm layer height.

## Casting of epi- and endothelial tubes

The protocol for casting tubes is as follows:

1. A 12 mm coverslip is attached to recessed area in the bottom of the TruD chip using preferably transparent nail varnish, as many types of colored nail varnish emit autofluorescence.

2. TruD chips are placed on a tray and sterilized by exposure to UV light in the TC hood for at least 15 minutes on all sides.

3. Next, the chips are transferred to a rack to facilitate handling during the steps below.

4. A 20/22-gauge syringe is inserted into the central pore of TruD chips.

5. 250 µl bovine collagen I is pipetted into central compartment with the needle inserted.

6. The needle is fully retracted and reinserted a couple of times. This facilitates coating of the parts of the pore that extends into the chip wall on either side of the central compartment.

7. The chip is placed in the TC incubator for approximately 120 minutes for the gel to solidify.

8. The needle is removed and 20-40-ul of $5x10^6$–$1x10^7$ epi-/endothelial cells/ml are injected with a 1-ml insulin syringe mounted with 27-gauge needle. To prevent the pore from expanding, it is important to thread the pore all the way to the distal end and inject the cells, while the syringe is being retracted at slow but uniform pace.

9. The TruD chips containing cells are placed in a 6-well dish and covered with culture medium (4 ml). The 6-well dish is then returned to the incubator for ~5 days in order for single layered tubes to form. At this point, tubes are quite stable and permit manipulation as described herein.

## Imaging

Four microscopes were used for imaging as follows:

1. Olympus TS-100 microscopy equipped with epifluorescence filters, THORLABS SOLOS-1C LED light source and DC20 controller, and Pixelink PL-D795MU monochrome Machine Vision Camera. Live and static images were sampled with PixeLINK Capture software. This microscope was used for live imaging of Telo-HAEC cells and for flow experiments with MDCK cells. For flow experiments, a 3D printed adaptor was designed to hold conjoined TruD chips and serve as a reservoir for excess fluid. Flow was generated and visualized by adding ~250 µl boluses of PBS+ containing 5% (v/v) red fluorescent microspheres to the reservoir during live imaging. Static images of Telo-HAEC cells were captured with a 4x objective. Live imaging of flow through an MDCK cell tube was recorded with 10x objective. For generating videos from stacks, images from z-series acquisitions were imported into Fiji (ImageJ) and exported as.avi files, which in turn were converted into.mp4 files.

2. Keyence BZ-X710 All-in-One-Fluorescence microscope was used for the bulk of fluorescence and brightfield imaging reported herein, both of live as well as fixed samples using 2X, 4X, 10X and 20X objectives. For lower magnifications, TruD chips were imaged while still contained inside a 6-well dish. For higher magnifications, the TruD chips were placed in 3D printed adaptor with the same exterior dimensions as proprietary Keyence adaptors (S9 and S11 Figs).

3. Zeiss LSM-980 confocal microscope equipped with the Airyscan 2 module was utilized to generate 3D reconstructions of MDCK cell tubes through a 20X/0.8NA objective. To this

end, chips containing tubes of MDCK cells expressing EGFP were fixed in 3% formalin for 15 minutes, rinsed/permeabilized three times with PBS containing 1% triton X-100, and stained with Alexa Fluor™ 594 Phalloidin diluted 1:100. Next, 2-channel fluorescent volumes of GFP positive and phalloidin stained tubes were acquired via Airyscan SR-8Y imaging mode with lasers and detectors tuned to fill 95% of the dynamic range in the brightest imaging plane. Data was processed with the Airyscan processing module and stitched in Zen Blue 3.6 from Zeiss Research Microscopy Solutions. 3D reconstructions were likewise generated with Zen Blue 3.6.

4. Applied Scientific Instrumentation (ASI) Dual Inverted selective plane illumination (diSPIM) lightsheet microscope. This microscope consists of a diSPIM "head" mounted onto a Nikon Eclipse TE2000-E microscope fitted with dual 40X Nikon CFI APO NIR Objectives (0.80 NA, 3.5 mm WD). The light source is a Toptica iChrome MLE laser with four lines (405/488/561/633 nm), while images are captured with high speed (100 fps) Hamamatsu Fusion ORCA-Fusion Digital CMOS cameras (C14440-20UP) using Aivia software for deconvolution. Lightsheet microscopy was performed on samples fixed in 3% formalin and stained with Alexa Fluor™ 594 Phalloidin as described above. Nuclei were moreover labeled with DAPI solution diluted 1:5000 (200 ng/ml final concentration). For imaging, the collagen gel containing tubes of MDCK cells expressing EGFP was removed from the TruD chip with fine tweezers and mounted on a standard microscope slide with nail varnish. In turn, the microscope slide was inserted into the ASI sample holder and the sample covered with PBS for immersion imaging. Video files of stacks were generates in Fiji (Image J) as described above.

## Results and discussion

### Casting tubes

The goal with developing TruD technology was to devise an inexpensive and user-friendly system for culturing epi- and endothelial cells in the shape of tubes, using generic culture vessels and lab supplies while permitting imaging on a wide range of microscopes. Moreover, although not included here, another goal is to permit recovery of tube cells for genetics and biochemistry, which will be reported at a later time. A chip design that fulfilled these criteria was is shown in Fig 1A. In brief, it is a 25-mm x 25-mm x 4-mm 3D printed chip that matches the width of a standard histology slide end hence mounts readily onto most microscope stages. It has recessed corners that are necessary for the chip to fit into a well of a 6-well tissue culture plate. The central area of the chip is a circular chamber for containing an ECM gel of choice. A recessed surface on the bottom of the chip permits mounting a standard 12-mm circular coverslip and thereby visualization of the tube that runs through the central compartment. Furthermore, there are cutouts to reduce buoyancy and thus prevent the chip from floating. Finally, the TruD chips are fitted with additional pores to potentially enhance diffusion of nutrients and permit for addition of cells, growth factors, etc. at defined angles relative to the tube. The latter feature is optional, while the others are essential. TruD chip designs and various support materials are visualized S1–S10 Figs and accompanied by 3D print files S1–S10 Files. Images of support materials are shown in S11 and S12 Figs.

A detailed protocol for casting tubes with TruD chips is included in the materials and methods section. In brief, a batch of chips are printed upside-down on a 3D printer using PLA+ filament with 100% infill and 0.15 mm layer height (Fig 1B). Next, 12 mm circular coverslips are mounted with one layer of nail varnish underneath the coverslip (Fig 1C and 1C'). After the

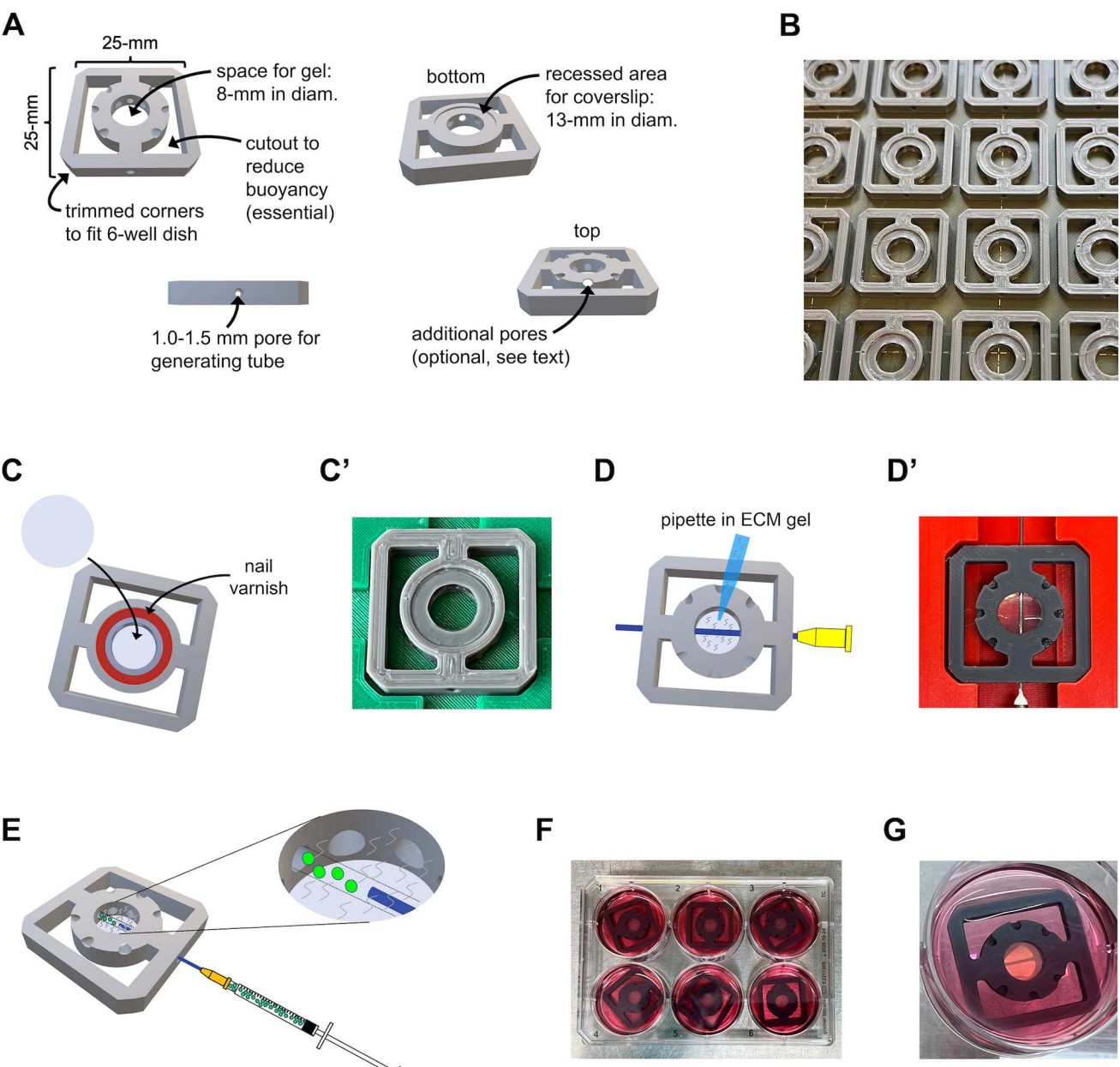

**Fig 1. Method for casting tubes with TruD chips.** (**A**) Basic chip design with description of features. (**B**) TruD chips on the bed of the 3D printer with the bottom facing up. (**C, C'**) Mounting of 12 mm coverslips in the recessed area on the bottom of the TruD chip. (**D, D'**) Pipetting ECM into the central chamber of the TruD chip after insertion of a 20/22-gauge needle into the central pore. (**E**) Cartoon illustrating the injection of cells of choice into the pore of the ECM generated by removal of the needle. (**F**) TruD chips with ECM gel and injected cells after transfer to 6-well dish containing 4-ml medium per well. (**G**) TruD chip in well of 6-well dish with cells injected into the central pore that are readily visible.

nail varnish has dried, the chips are placed on a tray in the TC hood and exposed to UV light for ≥ 15-min on each side. The chips are then transferred to a casting rack and a 20/22-gauge needle is inserted into the central pore (Fig 1D and 1D'). Next, 250 μl collagen gel is added to the central compartment each chip and the casting rack covered with a lid of a 6-well plate, before it is transferred to a 37°C incubator for two hours to permit the ECM to solidify.

Meanwhile, cells to be injected are harvested by trypsinization, pelleted and resuspended at a concentration of $5x10^6$–$1x10^7$cells/ml. After the gel is solidified, the needles are removed simply by holding the casting rack an ~60-degree angle, which is sufficient for the needles to fall out. The cells are then aspirated into a 1 ml syringe with a 27-gauge needle mounted, air bubbles removed, and 20–40 μl cell suspension injected into the pore of the gel (Fig 1E). Finally, the chips are transferred to 6-well dishes, covered with medium and placed in the incubator (Fig 1F). The time it takes for fully mature tubes to form will depend on the cell type, but it is approximately 5-days for the cell types described below. As shown in Fig 1G, the central pore containing injected cells is readily visible to the naked eye.

Others have described that multiple cell loadings of special handling were required to seed a porous lumen with cells [18]. In practice, this has not posed problems with TruD technology. With injection of a correct number of cells as described above, the pore wall is populated by cells within 24-hours. On occasion where insufficient cells are injected, the chip is discarded. A greater problem is posed by excess cells spilling into the 6-well dish and competing for media nutrients. This problem is addressed by transferring chips seeded with cells to a fresh 6-well dish.

## Simple tubes

Using the protocol summarized above, it is straightforward to reproducibly cast simple tubes. For the work described herein, MDCK cells and Telo-HAECs were derivatized to express EGFP, while TR-T10 and MDA-MB-231 cells were engineered to express mCherry. The use of fluorescent cells facilitates development and presentation of the technology. While convenient, the use of fluorescent cells may be undesired in some settings and is in fact entirely optional. Tubes can also be visualized by phase contrast imaging with or without vital stains. Fully formed tubes with distinct lumen and walls run through the entire length of the chip, as illustrated for MDCK cell tubes in Fig 2A–2C. Tubes may contain small "abnormalities" in the form of luminal protrusions (Fig 2D), but this also applies to cysts in ECM. It has not been possible to visualize the inside of the pore in the wall of the chip, but imagining immediately adjacent to the wall suggests that tubes extend into the chip itself (Fig 2E). It was next determined by confocal microscopy that the tubes are in fact uniformly hollow (Fig 2F), while lightsheet microscopy revealed that the tube wall is comprised by a single layer of cells (Fig 2G and 2G', and S1 Video). Staining with fluorescent phalloidin further demonstrated the presence of polymerized actin on the luminal face of MDCK cells consistent with the presence of a terminal web (Fig 2G and 2G', arrowheads). Intriguingly, imaging along the wall of the tube reveals cells with fragmented nuclei in the process of being extruded from the cell monolayer (Fig 2G, arrow), as previously described by others and ourselves for epithelial cells cultured on permeable supports [17,19].

To test the capacity of TruD technology to generate endothelial tubes, Telo-HAECs expressing EGFP were injected into the pore of collagen I gels. The protocol used to cast Telo-HAEC tubes is identical to the one described above for MDCK cells, and the tubes mature with similar kinetics (Fig 2H and 2H', and S2 Video). To demonstrate that Telo-HAEC cells form tubes, we injected human mammary carcinoma MDA-MD-231 cells expressing mCherry, which were retained within the lumen (Fig 2I and 2I'). This approach further suggests utility of TruD technology for studying processes such extravasation of cancer cells.

## Parallel tubes

To mimic more complex tube systems, as for instance vascularized tissue, it is desirable to cast tubes adjacent to one another. In turn, this would permit studies of complex biological

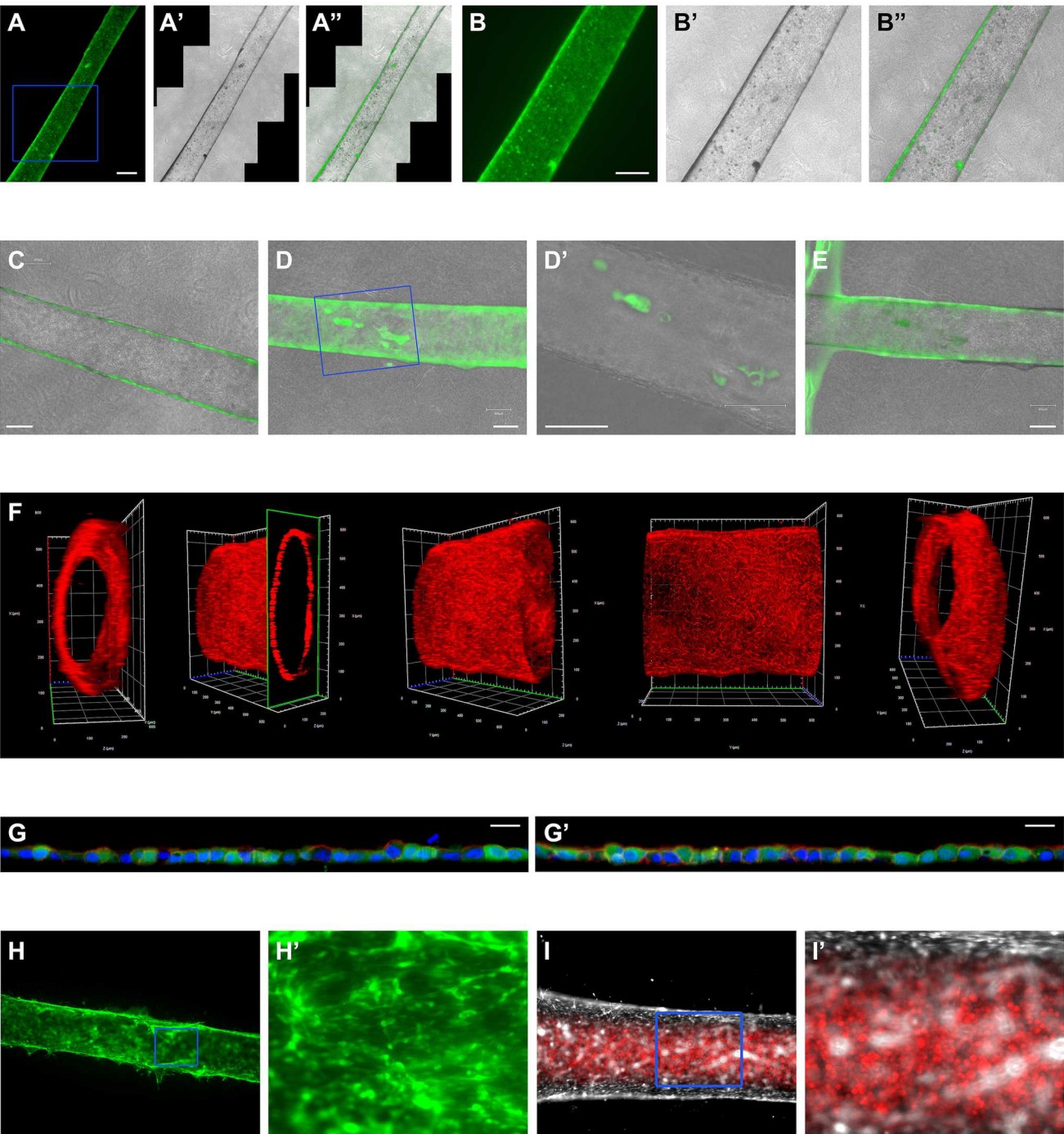

**Fig 2. Simple epi- and endothelial tubes cast with TruD chips.** (**A-A"**) MDCK cell tube cast with TruD chip visualized along the entire 8 mm length by fluorescence microscopy to detect EGFP (**A**), by brightfield microscopy (**A'**); and by overlay of fluorescence and brightfield microscopy (**A"**). Scalebar corresponds to 500 μm. (**B-B"**) Higher magnification of the area enclosed with in the blue rectangle shown in (**A-A"**). Scalebar corresponds to 500 μm. (**C, D, D'**) Representative examples of MDCK cell tubes cast with TruD chips. Note that the tube in (**C**) lacks "imperfections" that are visible in (**D, D'**). Scalebar corresponds to 250 μm. (**D'**) Higher magnification of the area enclosed with in the blue rectangle shown in (**D**) illustrates that the TruD chip technology can reveal small tube abnormalities. Scalebar corresponds to 250 μm. (**E**) Imaging along the perimeter of the central compartment of TruD chip shows that the MDCK cell tube minimally extends to the interface between ECM and the pore inside the wall of the chip. Scalebar corresponds to 250 μm. (**F**) 3D reconstruction of section of MDCK cell tube labeled with Alexa Fluor™ 594 Phalloidin (red) to detect polymerized actin. (**G, G'**) Lightsheet microscopy of wall of MDCK cell tube cast with TruD chip demonstrates that the wall of the tube consists of a single layer of cells. In this experiment, MDCK cells expressing EGFP (green) were fixed and stained with Alexa Fluor™ 594 Phalloidin (red) to reveal polymerized actin, which is enriched on the

lumenal surface (arrowheads). Moreover, the nuclei were labeled with DAPI (blue) of which one is extruded from the monolayer (arrow). Scalebars correspond to 10 μm. (**H, H'**) Tube cast with TeloHAEC cells expressing EGFP. The area enclosed by a blue rectangle in (**H**) is shown at higher magnification in (**H'**). Scalebars correspond to 200 μm (**H**) and 100 μm (**H'**), respectively. (**I, I'**) The lumen of tubes cast with TruD chips is readily accessible as shown here by injection on MDA-MB-231 mammary carcinoma cells expressing mCherry into a tube of TeloHAECs expressing EGFP, but rendered white using the screen effect in Adobe Photoshop to better distinguish the two cell types. Scalebars corresponds to 200 μm (**I**) and 50 μm (**I'**), respectively.

processes, as for instance neutrophil transmigration where neutrophils in response to infection exit the bloodstream and migrate across an epithelium with microbes in its lumen [20,21]. To this end, the TruD chip was modified to include parallel pores (Fig 3A). Using this design, it is feasible to cast tubes with different cell lines, as demonstrated in Fig 3B. For this figure, parallel tubes were cast with MDCK and TR-T10 cells expressing mEGFP and mCherry, respectively. Evidently, complexities multiply when two tubes are cast in close proximity of one another. The problems include (i) preventing spill-over of cells between adjacent tubes; (ii) tearing of the gel when two pores need to be threaded successively; and (iii) the distance between tubes complicating simultaneous imaging of parallel tubes at higher resolution. However, once cast, parallel tubes do not appear less stable than simple tubes.

Fig 3B illustrates another important aspect of casting tubes with the technology described here. Using 20/22-gauge needles should create pores and hence tubes with 900/700 μm lumens. In practice, however, the tubes are significantly smaller and the diameter is cell line-dependent. Thus, despite highly standardized conditions, the actual tube diameter achieved with a given cell line is influenced by both cell-extrinsic and -intrinsic factors.

## Serial tubes

Many tubes in our body are specialized along the length. Obvious examples include the intestine, kidneys and exocrine glands. To model such tubes, it is desirable to be able to join TruD chips with tubes containing different types of cells. To this end, a modular design has proven successful (Fig 3C). One chip–male–is printed with a tubular extension that fits into a recessed pore in the second–female–chip. To demonstrate utility of this design, male and female TruD chips were seeded separately with green and red fluorescent MDCK and TR-T10 cells, respectively, and propagated until tubes were fully formed on day 5 or later (Fig 3D). Then, male and female TruD chips were "clicked" together in a 100 mm dish containing PBS+. Next, to test whether contents can flow from one tube to another, we injected 20 μl of blue and red fluorescent microspheres into the free orifice of the green tube contained within male chip (Fig 3E). After imaging of the green tube within the male chip (Fig 3F), 20 μl of PBS was pushed into the proximal end of the male chip, and the female chip was then imaged. Under these conditions, it was determined that, the lumen of the female tube, but not its surroundings, indeed contained red and blue microspheres inside (Fig 3G). The only manner in which the microspheres could reach the tube inside the female chip was via flow from the tube in the male chip. The setup along with magnified areas of the connected male and female tubes are shown in Fig 3H and 3I, respectively. This result demonstrates that the lumen of the green and red tubes in the male and female TruD chips, respectively, are connected. The modular design for generating serial tubes should be useful for a variety of biological applications. For instance, one could generate a mini-gut by seeding small and large intestinal enteroids in the male and female chip, respectively. Likewise, it should prove useful for adjoining proximal and distal tubule cells from the kidney with numerous other applications imaginable.

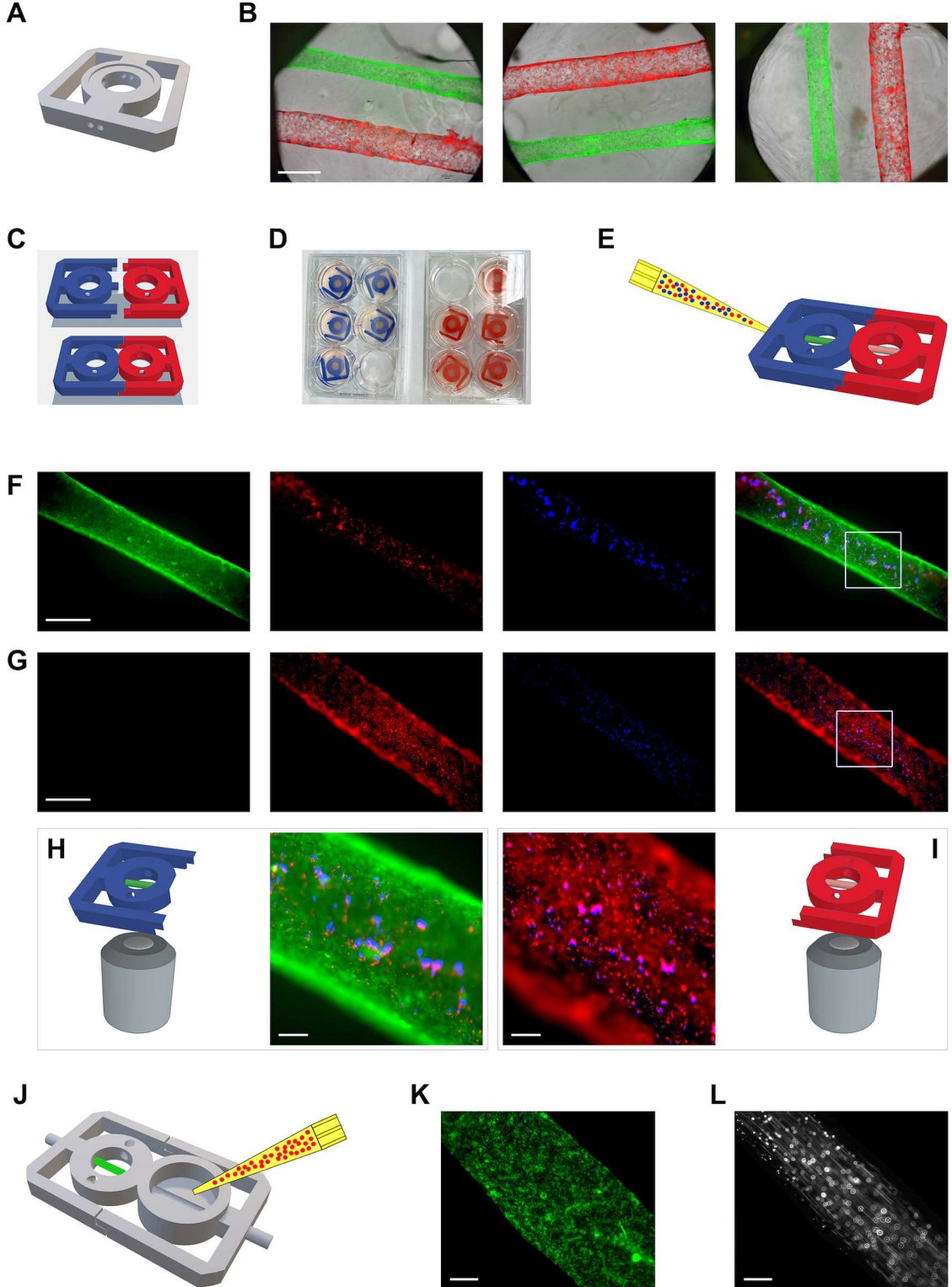

**Fig 3. Parallel and serial tubes, as well as tubes with flow cast with modified TruD chips.** (**A**) Modified TruD chip to accommodate two parallel tubes. (**B**) Examples of parallel tubes cast with TruD chips. Cells in the green and red tubes express EGFP and mCherry, respectively. Scalebar corresponds to 1 mm. (**C**) Model depicting male (blue) and female (red) TruD chips, as well as how they are conjoined. (**D**) Culture of EGFP- and mCherry-expressing MDCK and TR-T10 cells in male (blue) and female (red) chips, respectively. (**E**) Model illustrating injection of blue- and red-fluorescent microspheres into the male pore opening of conjoined

chips. (**F**) Imaging of the male, EGFP-expressing, MDCK cell tube after injection of microspheres into the male chip. Scalebar corresponds to 300 μm. (**G**) Imaging of the female, mCherry-expressing, TR-T10 cell tube after injection of 25 μl PBS into the male chip. Scalebar corresponds to 300 μm. (**H**) Magnification of the area enclosed by a white box in (**F**). Scalebar corresponds to 50 μm. (**I**) Magnification of the area enclosed by a white box in (**G**). Scalebar corresponds to 50 μm. (**J**) Model illustrating setup for flow experiment using a reservoir clipped together with a male chip. (**K**) Visualization of a tube segment of EGFP-expressing MDCK cells immediately prior to addition of solution containing microspheres to the attached reservoir. Scalebar corresponds to 100 μm. (**L**) Flow of microspheres through the tube segment shown in (**k**). Scalebar corresponds to 100 μm.

## Tubes with flow

A defining capacity of tubes is that they sustain flow, a feature that is not replicated by culturing epi-/endothelial cells on permeable supports or as cysts immersed in ECM. To achieve flow, TruD chips were modified to include small cylindrical extensions on either side that permit attachment of tubing and a microfluidic pump if desired (Fig 3J). The internal and external diameters of the extensions measure 1.75 mm and 2.45 mm, respectively to permit attachment of generic peristaltic pump tubing and potentially incorporate Luer fittings. Furthermore, the male-female chip design was modified to include a reservoir in the female chip and thereby permit hydrostatic pressure to drive flow (Fig 3J). The open reservoir also serves as an essential function in mitigating back-pressure, the importance of which cannot be overstated. After culturing green MDCK cell tubes in the male TruD chip, a reservoir was attached as described above. Next, after imaging the green tube (Fig 3K) boluses of ~250 μl PBS+ containing fluorescent beads were pipetted into the reservoir and flow through the tube was visualized by live imaging (Fig 3L, S3 and S4 Videos). This was repeated four times with five-minute intervals with the tube remaining fully intact.

## Conclusions

This work describes a simple suite of devices for generating various epi- and endothelial tubes in vitro using nothing more than an inexpensive 3D printer and common lab supplies. The simplicity and modular nature is what distinguishes TruD technology described herein from previously reported more elaborate methods for generating epi- and endothelial tubes [5–16]. Hence, TruD technology is accessible for users without bioengineering expertise. It allows for experiments to be conducted on the same scale as conventional culture on permeable supports and in 3D matrices. Moreover, the modular design permits mimicking of physiological structures. Finally, TruD technology is pliable in a manner that allows the end user to adapt the design to suit individual needs. To this end, numerous refinements are in the works to manipulate tube size by using finer needles for casting and injection, modify the ECM content, introduce stromal and immune cells, as well as a modified chip designs to reduce consumption of ECM and culture media. Such efforts will be facilitated by using dual nozzle 3D printers that permit printing of support material combined with a smaller nozzle size, 0.25 μm vs 0.4 μm. However, until such refinements have been successfully achieved, they represent potential limitations. Extensive validation is necessary to establish utility of TruD technology to physiological and pathobiological areas as exemplified by the following: (i) epi-/endothelial tube interactions with stromal cells and components; (ii) bi-layered tubes to mimic, for instance, epithelial ducts with myoepithelial cells in the external layer; (iii) paracrine signaling; (iv) transepi- and endothelial transport; (v) intra- and extravasation; (vi) secretory disorders; (vi) oncogenic transformation, etc. Hopefully other laboratories will find TruD technology of interest to their respective areas of expertise and participate in further development.

## Supporting information

**S1 Fig. Basic TruD chip.**
(PDF)

**S2 Fig. TruD chip with parallel pores.**
(PDF)

**S3 Fig. Male TruD chip.** Note, the print support below the tubular extension is removed prior to usage.
(PDF)

**S4 Fig. Female TruD chip.** Note, the print supports beneath the external walls are removed prior to usage.
(PDF)

**S5 Fig. Female chip and male TruD chips of which the latter is modified with a reservoir to permit flow.** Print supports, as described in legends to S3 and S4 Figs, are removed prior to usage.
(PDF)

**S6 Fig. Tray for handling chips while mounting coverslips and for storage prior to usage.**
(PDF)

**S7 Fig. Rack for facilitating handling of TruD chips during pipetting of ECM as well as injection of cells.** The dimensions of the rack match the bottom of a 6-well dish such that the lids can be used to cover the chips to avoid contamination while the ECM solidifies after placement in the incubator.
(PDF)

**S8 Fig. Alphanumeric holder for enabling easy transport and short-term storage of TruD chips.**
(PDF)

**S9 Fig. TruD adaptor for Keyence microscope.**
(PDF)

**S10 Fig. Cabinet for dust-free storage of TruD chips.**
(PDF)

**S11 Fig. Examples of 3D prints of the.stl files illustrated in S6–S9 Figs.** (**A**) Tray for mounting coverslips and storage. (**B**) Casting rack. (**C**) Alphanumeric holder. (**D**) Adaptor for Keyence microscope.
(PDF)

**S12 Fig. 3D print of the cabinet 10.stl file shown in S10 Fig.**
(PDF)

**S1 File. TruD chip basic.**
(STL)

**S2 File. TruD chip parallel.**
(STL)

**S3 File. TruD chip male.**
(STL)

**S4 File. TruD chip female.**
(STL)

**S5 File. TruD chip reservoir.**
(STL)

**S6 File. TruD tray.**
(STL)

**S7 File. TruD rack.**
(STL)

**S8 File. TruD holder.**
(STL)

**S9 File. TruD Keyence adaptor.**
(STL)

**S10 File. TruD cabinet.**
(STL)

**S1 Video. Stack of wall of GFP expressing MDCK cell tube imaged by lightsheet fluorescence microscopy.** Red stain: DAPI. Green stain: GFP. Blue: Alexa-phalloidin[594]. Note that blue and red colors are transposed from Fig 1G and 1G'.
(MP4)

**S2 Video. Stack of GFP expressing Telo-HAEC tube imaged by fluorescence microscopy.**
(MP4)

**S3 Video. Flow of red fluorescent microspheres through MDCK cell tube at imaged t = 0 mins.**
(MP4)

**S4 Video. Flow of red fluorescent microspheres through MDCK cell tube imaged at t = 15 mins.**
(MP4)

## Acknowledgments

I am most grateful to Raj Kampka (BCH) for productive discussions. I am also highly appreciative of help with imaging on various microscopes from Maksymilian Prondzynski (BCH), Jay Thiagarajah (BCH) and Harry Kramer at the IDDRC Cellular Imaging Core (BCH), as well as to Matthew P. Vivero and Patrick J Smits for Telo-HAEC cells. This work is dedicated to Steen A. Urbin-Hansen for his keen interest in the project.

## Author Contributions

**Conceptualization:** Steen H. Hansen.

**Data curation:** Steen H. Hansen.

**Formal analysis:** Steen H. Hansen.

**Funding acquisition:** Steen H. Hansen.

**Investigation:** Steen H. Hansen.

**Methodology:** Steen H. Hansen.

**Project administration:** Steen H. Hansen.

**Resources:** Steen H. Hansen.

**Supervision:** Steen H. Hansen.

**Validation:** Steen H. Hansen.

**Visualization:** Steen H. Hansen.

**Writing – original draft:** Steen H. Hansen.

**Writing – review & editing:** Steen H. Hansen.

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
