## [Decision Letter · Decision Letter 0]

8 Feb 2024

PONE-D-24-00292TruD technology for the study of epi- and endothelial tubes in vitroPLOS ONE

Dear Dr. Hansen,

Thank you for submitting your manuscript to PLOS ONE. After careful consideration, we feel that it has merit but does not fully meet PLOS ONE’s publication criteria as it currently stands. Therefore, we invite you to submit a revised version of the manuscript that addresses the points raised during the review process.

We look forward to receiving your revised manuscript.

Kind regards,

Jérôme Robert, PhD

Academic Editor

PLOS ONE

Journal Requirements:

"ACKNOWLEDGEMENTS

I am most grateful to Raj Kampka for productive discussions. I am also highly appreciative of help with imaging

on various microscopes from Maksymilian Prondzynski, BCH Division of Cardiology, Harry Kramer at the

IDDRC Cellular Imaging Core at BCH for the use of the LSM 980 system; funded by NIH S10 OD030322

& NIH P50 HD105351, and Jay Thiagarajah at the Harvard Digestive Diseases Imaging Core; sponsored by

P30 DK034854. This work was supported by an endowed chair from the Roy and Lynne Frank Foundation to

S.H.H."

**Additional Editor Comments:**

I suggest that the author answers the questions of the reviewer 2 and simply discusses the limitations raised by the reviewer 1 in the discusssion. 

Reviewers' comments:

Reviewer's Responses to Questions

**Comments to the Author**

1. Is the manuscript technically sound, and do the data support the conclusions?

Reviewer #1: Partly

Reviewer #2: Yes

2. Has the statistical analysis been performed appropriately and rigorously? 

Reviewer #1: N/A

Reviewer #2: N/A

3. Have the authors made all data underlying the findings in their manuscript fully available?

Reviewer #1: No

Reviewer #2: Yes

4. Is the manuscript presented in an intelligible fashion and written in standard English?

Reviewer #1: Yes

Reviewer #2: Yes

5. Review Comments to the Author

Reviewer #1: The authors present a new 3D model that can generate tubular structures similar to the ones found in human organs. They demonstrate the potential of this model for various applications, such as blood vessels, kidney tubes and intestines. I recommend the authors to perform more validations of the 3D model performing experiments that help to analyse the reproducibility of the model.

The process of extravasation, which is the movement of cells from the blood vessels to the surrounding tissues is one of the aspects proposed to evaluate with this 3D model . In effect, the model is innovative and intriguing, but it requires more validation and quantification to demonstrate its applicability and reliability in the field of extravasation research. For instance, one possible experiment that could be done is to use monocytes that are marked with a fluorescent dye and track their extravasation after stimulating the endothelial cells with pro-inflammatory cytokines. This would serve as a positive control and could be compared with a negative control where the endothelial cells are not stimulated. This way, the model could show how well it can reproduce the physiological conditions and measure the extravasation rate and efficiency.

The auteurs also demonstrate that the model is able to create a lumen where they can inject molecules and create a flow. However this aspect could be improved using a pulsatile fluid flow instead of just molecule injection. It would be interesting to see how the model behaves under different flow regimes and whether this affects the extravasation process or not for example. This would also help to evaluate the robustness and sensitivity of the model to different parameters and inputs.

Reviewer #2: This manuscript presents a way to create a cylindrical lumen inside a hydrogel that can subsequently be seeded with cells, and the cylindrical channels can be perfused. The ideas presented in this work build on numerous examples of previous work in the literature that have demonstrated casting of patterned vascular networks (e.g. Miller et al. 2012, https://www.nature.com/articles/nmat3357), patterning of luminal structures in ECM gels (e.g. Jiménez-Torres et al. 2016, https://doi.org/10.1002%2Fadhm.201500608), among other examples. The strengths of this present work include sharing the .stl files such that other researchers will be able to create the 3D printed chips; the components are readily, commercially available.

The present work uses 20/22-gauge needles to mold the lumen, thus creating approximately 900 um to 700 um diameter lumens. The use of commercially available needles may become challenging as the need for smaller diameter lumens may arise in some applications, such as modeling of microvasculature or capillaries with diameters on the order of tens of microns.

The method presented here appears to have allowed the seeding of cells inside the relatively large diameter lumen. Other work in the literature shows that multiple cell loadings and/or a combination with device rotation, either with a motor or by manual rotation (e.g. Bischel et al. 2012, https://doi.org/10.1016%2Fj.biomaterials.2012.11.005), is needed to assure cells attach evenly over the interior of the lumen. The author may wish to comment on whether additional techniques are needed to ensure that cells will adhere evenly over the relatively large diameter lumen created in this work.

The author may also wish to comment on the ease of connecting these chips to perfusion systems for controlled, continuous perfusion over longer culture periods. While the current design permits nutrients to diffuse through the top surface of the hydrogel, and the TruD chips have “small extensions on either side that permit attachment of tubing and a microfluidic pump if desired”, the author may wish to comment on how future iterations of the design may potentially incorporate industry-standard interconnects, such as luer locks, which can facilitate easier assembly and connection to pumps for long-term perfusion of the lumen. The long-term perfusion may also be needed to re-create physiological shear stress conditions at the surface of the cell layers.

It is not yet clear why the pores (shown in Fig. 1A) in the 3D printed structure would be needed for diffusion of nutrients to the gel. From this design, it appears that the bottom of the chamber is defined by the circular coverslip, and that this confines the bottom surface of the ECM. It appears that the top of the ECM gel is open to the culture media, and that nutrients would diffuse into the lumens or tubes through the ECM. It is not clear that the diffusion path through the pores in the frame of the chip are needed for delivering nutrients to the gel, as those pores appear to be further from the cells inside the gel than the distance from the top of the gel to the tube formed within the gel. The author may wish to comment on this design feature.

Overall, this chip design appears to work for the cells and the gel that was used here. Users may encounter challenges with different cell types that may contract the gel (e.g. fibroblasts contracting the collagen in a co-culture model with endothelial cells lining the lumen); practical challenges in connecting the chips for controlled, long-term perfusion to create physiological shear stress; challenges with simply imaging in brightfield (without fluorescent cells) in assessing whether the seeded cells have formed a confluent monolayer, as imaging through the mm-thick gel in this 3D construct is more challenging than imaging monolayers on flat substrates; other practical challenges as they arise. These will be determined by the specific applications of each user.

6. PLOS authors have the option to publish the peer review history of their article (what does this mean?). If published, this will include your full peer review and any attached files.

Reviewer #1: No

Reviewer #2: No

---

## [Author Response · Author response to Decision Letter 0]

8 Mar 2024

Response to Reviewers of PONE-D-24-00292

I am most grateful to the reviewers for their insightful and constructive critiques of the initial submission of PONE-D-24-00292. Below, I address their comments point-by-point. Specific reviewer comments, as I perceive them, have been italicized. My rebuttal is written with blue text for ease of distinction.

Reviewer #1

1. “I recommend the authors to perform more validations of the 3D model performing experiments that help to analyse the reproducibility of the model… In effect, the model is innovative and intriguing, but it requires more validation and quantification to demonstrate its applicability and reliability in the field of extravasation research. For instance, one possible experiment that could be done is to use monocytes that are marked with a fluorescent dye and track their extravasation after stimulating the endothelial cells with pro-inflammatory cytokines.”

I completely agree with these comments in principle. However, in practice, it is but one example along with an array of challenges that the TruD system should be put to in order to define its strengths and limitations. I made some attempts along these lines, but quickly realized that it requires more resources and manpower, which I presently do not possess. Instead, I came to the conclusion that a primer on the system that allows others to adapt the system to their needs would be more suitable. To this end, users will have free access to the .stl files, and I am happy collaborate to modify designs to individual needs. The discussion of the revised manuscript has been modified accordingly. Specifically, the following text has been added to the end of the discussion to acknowledge ths critique of Reviewer #1: “However, until such refinements have been successfully made, they represent potential limitations. Extensive validation is necessary to establish utility of TruD technology to physiological and pathobiological areas as exemplified by the following: (i) epi-/endothelial tube interactions with stromal cells and components; (ii) bi-layered tubes to mimic, for instance, epithelial ducts with myoepithelial cells in the external layer; (iii) paracrine signaling; (iv) transepi- and endothelial transport; (v) intra- and extravasation; (vi) secretory disorders; (vi) oncogenic transformation, etc. Hopefully other laboratories will find TruD technology of interest to their respective areas of expertise and participate in further development.”

2. “…this aspect (i.e. flow) could be improved using a pulsatile fluid flow instead of just molecule injection.”

I am bit confused by this comment. In the flow experiment illustrated in Figures 3 J-K and accompanying Videos 3&4 a bolus is added to the reservoir, which from my perspective results in a pulsative flow. In Video 3, the flow even reverses at the end. I have two additional videos from the same imaging session that I can add if the Reviewers and Editors so desire.

Reviewer #2

1. The ideas presented in this work build on numerous examples of previous work in the literature that have demonstrated casting of patterned vascular networks (e.g. Miller et al. 2012, https://www.nature.com/articles/nmat3357), patterning of luminal structures in ECM gels (e.g. Jiménez-Torres et al. 2016, https://doi.org/10.1002%2Fadhm.201500608), among other examples.

I am grateful for the additional literature references, of which I was unaware, but they have been cited in the revised manuscript to give the authors of these papers and their inventions proper credit.

2. The present work uses 20/22-gauge needles to mold the lumen, thus creating approximately 900 um to 700 um diameter lumens. The use of commercially available needles may become challenging as the need for smaller diameter lumens may arise in some applications, such as modeling of microvasculature or capillaries with diameters on the order of tens of microns.

In actuality, the lumens are smaller than the gauge of the needle to an extent that depends on the cell type or even clone. This is particularly evident from Figure 3B. It appears that the cells during formation of the tube contract the ECM in such manner that the actual tube diameter is smaller. However, the point raised by Reviewer #2 is completely valid and represents a limitation of the TruD technology in its current form. Again, there is a tradeoff between ease of use and accessibility and the capacities of the system. To cast smaller tubes, I am planning to use acupuncture needles combined with a modified chip design that allows for seeding of the tube volume by capillary effect, gravity, gentle vacuum or other means. In addition, it should be feasible to cast TruD chips with smaller pores by reducing the nozzle size from 0.4 µm to 0.2 µm and using a dual nozzle 3D printer to enable printing of removable support material. In its present design, the pores at minimum, below which the pores become oblong or collapse altogether. Moreover, the walls of the tubular extensions of the chips designed for flow studies are at a minimum. Along with other modifications alluded to in this rebuttal, substantial additional effort will be required by ourselves and hopefully others, but if successful, such will have to be included in future studies.

I have made revisions to the manuscript addressing these points in the revised manuscript. First the description of simple tubes now includes a brief paragraph to this effect: "Fig. 3B illustrates another important aspect of casting tubes with the technology described here. Using 20/22-gauge needles should create pores and hence tubes with 900/700 µm lumens. In practice, however, the tubes are significantly smaller and the diameter is cell line-dependent. Thus, despite highly standardized conditions, the actual tube diameter achieved with a given cell line is influenced by both cell-extrinsic and -intrinsic factors.”. Second, in the discussion now includes the following statement “…numerous refinements are in the works to manipulate tube size by using finer needles for casting and injection, modify the ECM content, introduce stromal and immune cells, as well as a modified chip designs to reduce consumption of ECM and culture media. Such efforts will be facilitated by using dual nozzle 3D printers that permit printing of support material combined with a smaller nozzle size, 0.25 µm vs 0.4 µm.”

3. “The author may wish to comment on whether additional techniques are needed to ensure that cells will adhere evenly over the relatively large diameter lumen created in this work.”

Surprisingly or not, this is a non-issue with the TruD chips. With the protocol described, most tubes will have the entire surface populated by cells within 24 hours. If there are smaller patches devoid of cells inside the pore, they are swiftly populated by cells. If insufficient cells accidentally are injected into the pore, the chip is simply discarded, because the cost is minimal with Collagen I. It might be a different matter with Matrigel or other expensive gel materials, but it is fairly easy to build up a routine where most chips are injected with the correct number of cells. A statement to this effect has been added to the revised manuscript as follows: "Others have described that multiple cell loadings of special handling were required to seed a porous lumen with cells [17]. In practice, this has not posed problems with TruD technology. With injection of a correct number of cells as described above, the pore wall is populated by cells within 24-hours. On occasion where insufficient cells are injected, the chip is simply discarded. A greater problem is posed by excess cells spilling into the 6-well dish and competing for media nutrients. This problem is addressed by simply transferring chips seeded with cells to a fresh 6-well dish."

4. “…the author may wish to comment on how future iterations of the design may potentially incorporate industry-standard interconnects, such as luer locks, which can facilitate easier assembly and connection to pumps for long-term perfusion of the lumen.”

This point has been taken into consideration and should have been mentioned in the original submission. Whether it is possible to print chips Luer locks, remains to be tested, but it is certainly possible to insert Luer fittings into tubing that connects to the extensions of TruD chips designed to accommodate flow. This is briefly discussed in the revised manuscript: "The internal and external diameters of the extensions measure 1.75 mm and 2.45 mm, respectively to permit attachment of generic peristaltic pump tubing and potentially incorporate Luer fittings."

5. “It is not yet clear why the pores (shown in Fig. 1A) in the 3D printed structure would be needed for diffusion of nutrients to the gel. From this design, it appears that the bottom of the chamber is defined by the circular coverslip, and that this confines the bottom surface of the ECM. It appears that the top of the ECM gel is open to the culture media, and that nutrients would diffuse into the lumens or tubes through the ECM. It is not clear that the diffusion path through the pores in the frame of the chip are needed for delivering nutrients to the gel, as those pores appear to be further from the cells inside the gel than the distance from the top of the gel to the tube formed within the gel. The author may wish to comment on this design feature.”

Point taken. This was rather clumsily presented and perhaps overstated on my part in the original submission. Presently, I do not possess any data to document that these additional pores are necessary for viability of cells inside tubes of TruD chips. However, the pores do reach the central compartment of the chips at a distance of ~1-mm from the coverslip. Hypothetically, they should facilitate diffusion of at least macromolecules into the bottom part of the chamber. Also, the permit injection of cells, antibodies, proteins etc. at defined angles relative to the tube, which might provide useful for polarity studies. Accordingly, the text of the revised manuscript has been modified to read: "Finally, the TruD chips are fitted with additional pores to potentially enhance diffusion of nutrients and permit for addition of cells, growth factors, etc. at defined angles relative to the tube." Furthermore, Figure 1A has been modified, to avoid overstating the utility of these extra pores.

---

## [Editor Report · Decision Letter 1]

12 Mar 2024

TruD technology for the study of epi- and endothelial tubes in vitro

PONE-D-24-00292R1

Dear Dr. Hansen,

We’re pleased to inform you that your manuscript has been judged scientifically suitable for publication and will be formally accepted for publication once it meets all outstanding technical requirements.

Kind regards,

Jérôme Robert, PhD

Academic Editor

PLOS ONE
---

## [Editor Report · Acceptance letter]

29 Apr 2024

PONE-D-24-00292R1 

PLOS ONE

Dear Dr. Hansen, 

I'm pleased to inform you that your manuscript has been deemed suitable for publication in PLOS ONE. Congratulations! Your manuscript is now being handed over to our production team.

Kind regards, 

on behalf of

Dr. Jérôme Robert 

Academic Editor

PLOS ONE